# Direct access to poly(glycidyl azide) and its copolymers through anionic (co-)polymerization of glycidyl azide

Senthil K. Boopathi[1], Nikos Hadjichristidis[2], Yves Gnanou[1] & Xiaoshuang Feng[1]

Glycidyl azide polymer or poly(glycidyl azide) which is considered as an excellent energetic binder or plasticizer in advanced solid propellants is generally obtained by post-modification or azidation of poly(epichlorohydrin). Here we report that glycidyl azide can be directly homopolymerized through anionic ring-opening polymerization to access poly(glycidyl azide) using onium salts as initiator and triethyl borane as activator. Molar masses of poly(glycidyl azide) up to 11.0 Kg/mol are achieved in a controlled manner with a narrow polydispersity index (PDI ≤ 1.2). Similarly, alternating poly(glycidyl azide carbonate) are also prepared through alternating copolymerization of glycidyl azide with carbon dioxide. Lastly, the copolymerization of glycidyl azide with other epoxide monomers is carried out; the azido functions carried by glycidyl azide which are successfully incorporated into the backbones of polyethers and polycarbonates based on cyclohexene oxide and propylene oxide subsequently served to introduce other functions by click chemistry.

[1] Physical Science and Engineering Division, King Abdullah University of Science and Technology (KAUST), Thuwal 23955, Saudi Arabia. [2] KAUST Catalysis Center, Physical Science and Engineering Division, King Abdullah University of Science and Technology (KAUST), Thuwal 23955, Saudi Arabia. Correspondence and requests for materials should be addressed to Y.G. (email: yves.gnanou@kaust.edu.sa) or to X.F. (email: fxs101@gmail.com)

Glycidyl azide polymer or poly(glycidyl azide) (PGA) is considered as excellent energetic binder or plasticizer in advanced solid propellants because of its high heat of combustion, thermal stability, and good compatibility with oxidizers[1-4]. Di- or trihydroxyl terminated PGAs ($M_n = 1000-4000$) are conventionally reacted with di- or triisocyanates together with oxidizers and other additives to form polyurethane networks, the pendent explosophore azido groups in PGA serving to improve the energetics of the binder and boost the overall performance of propellants. PGA can also react with alkynes through its azido functions to form 1,2,3-triazoles by click chemistry which is an alternate route to access propellant binders with high energies[5,6].

No literature reference could be found indicating how to polymerize glycidyl azide (GA) to obtain the corresponding polymer, PGA. For the applications mentioned above the only described methods were those resorting to the post-modification or azidation of poly(epichlorohydrin) (PECH) to generate PGA[7]. For instance, Vandenberg described such a modification to obtain PGA through the example of PECH-triol, using $NaN_3$ for the substitution reaction in dimethyl sulfoxide (DMSO)[8]. Greener azidation processes, involving solvents such as ionic liquids[9] and water together with phase transfer catalysts[1,4] were later reported to address environmental concerns related to the substitution reaction.

In contrast to the prior literature, we report in this paper how PGA can be directly obtained from the anionic ring-opening polymerization (AROP) of GA in the presence of a mild Lewis acid, triethyl borane (TEB). We discuss the role of TEB which is essential for the successful synthesis of PGA. In the presence of this particular Lewis acid, the AROP of GA occurs in a controlled manner in a range of molar masses between 1000 and 11,000 g/mol. α,ω-dihydroxy telechelics of very low polydispersity carrying all of their azido functions and thus responding to the requirement of propellant binder can be easily prepared. Besides the homopolymerization of GA, its copolymerization with other monomers is also investigated. Its copolymerization with $CO_2$ which results in the synthesis of poly(glycidyl azide carbonate) (PGAC) is meant to endow degradability to PGA[10]. Moreover GA-based random or block copolymers are derived upon copolymerization of GA with other epoxide monomers. Finally, PGA is shown to serve as the precursor for the synthesis of other functional polyethers through the derivatization of its azido groups through "click" reactions or reduction into primary amines[7].

## Results

**Homopolymerization of GA.** Since GA is not amenable to either pure anionic or cationic initiation or ring opening polymerization due to the presence of its reactive[11] and acid sensitive azido function[12,13] we resorted to the strategy of ate complex to initiate its polymerization. Such ate complexes are formed upon mixing anions with Lewis acids (Supplementary Figure 1). We could for instance successfully copolymerize epoxides with $CO_2$ by initiating the polymerization with a boron-ate complex and therefore avoid the formation of cyclic carbonates which would be the outcome of a pure anionic copolymerization. In the presence of TEB and a salt used as initiator poly(cyclohexene carbonate) (PCHC) and poly(propylene carbonate) (PPC) could be indeed obtained under "living" conditions[10]. In the case investigated here a boron-ate complex derived from TEB and an anionic initiator also helped to successfully polymerize GA, provided TEB was added in slight excess.

The first attempt was actually carried out without any excess of TEB in the presence uniquely of the boron-ate complex stoichiometrically formed by mixing TEB and the anionic initiator; under such conditions the polymerization was sluggish whatever the temperature tried: 6% conversion at room temperature after 18 h, 14% conversion at 40 °C after 18 h, 32% conversion at 40 °C after 40 h. We then decided to use an excess of TEB to activate specifically the monomer GA, a strategy we utilized previously for the copolymerization of $CO_2$ with epoxides[10]. TEB has thus a dual role as shown in Fig. 1. It serves first to form an ate complex upon its mixing with the initiating and then propagating anion; such ate complexes being less nucleophilic than bare anions transfer reactions are minimized and only propagation occurs, provided an excess of TEB is added to activate the monomer: this is the second role of TEB (See Fig. 2). The characterization by NMR of the GA/TEB mixture clearly shows that for a 3 equivalent excess of TEB a slight downfield shift of the epoxide protons due to the activation of the epoxide by TEB (Fig. 2b and inset). A first experiment carried out at 40 °C (entry 1, Table 1) using $Bu_4NBr$ as initiator

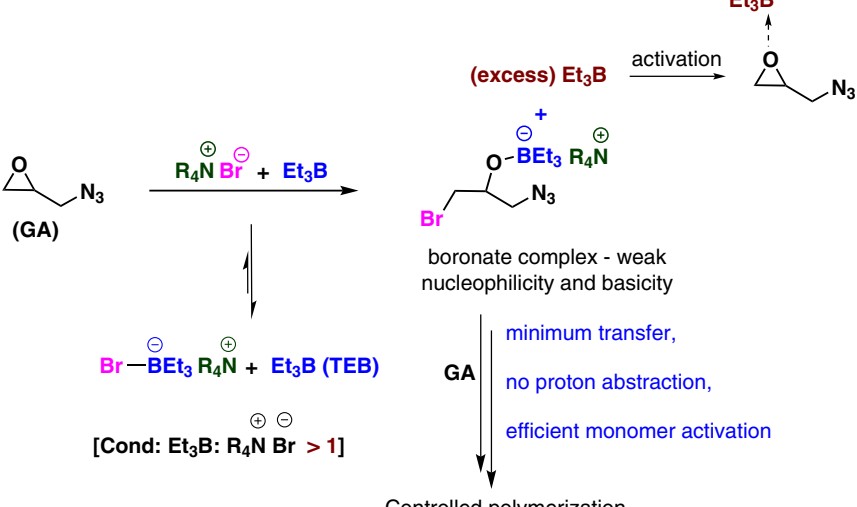

**Fig. 1** Role of excess of TEB in GA Polymerization. An excess of TEB is required for the GA polymerization to occur. Firstly, an ate complex is formed upon mixing stoichiometrically TEB with the initiating anion; such ate-complexes exhibit a weak nucleophilicity and basicity which prevents the occurrence of undesired transfer reactions and proton abstraction from the monomer. Secondly, the excess of TEB serves to activate the monomer

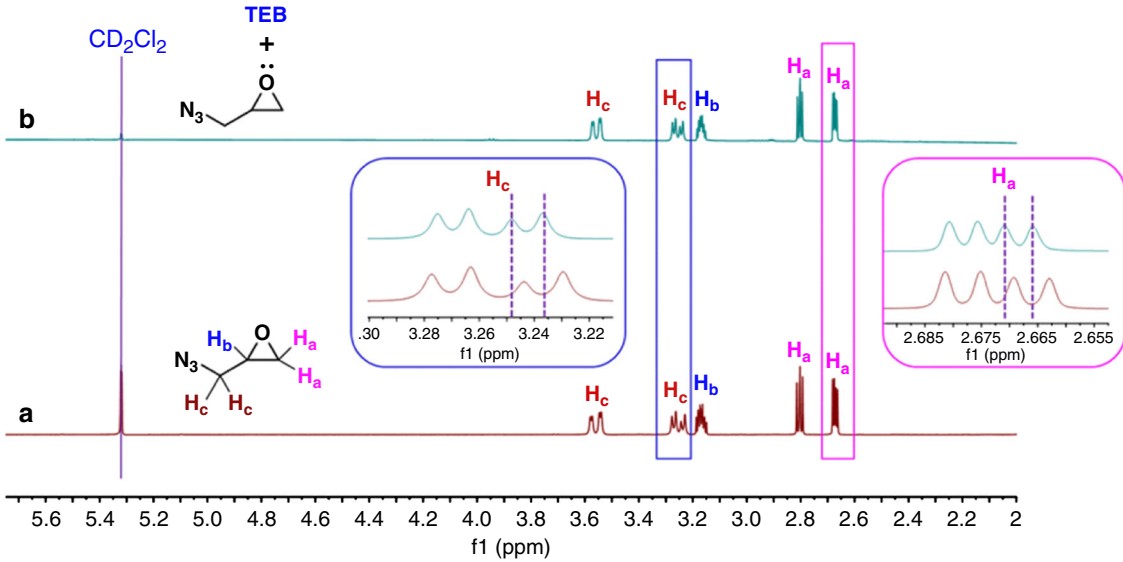

**Fig. 2** $^1$H NMR spectra showing the activation of GA by TEB. **a** Shows the $^1$H NMR spectrum of GA. **b** Displays the interaction of TEB with the epoxide and namely with its azido group as indicates the downfield shift of the methylene protons of the epoxide (H$_a$) and of its azido group (H$_c$)

**Table 1 Triethyl borane-assisted homopolymerization of glycidyl azide$^a$**

| Entry | Initiator | [M]:[I]:[TEB] | Time (h) | Temp ($^O$C) | Conv (%)$^b$ | M$_{n(theo)}$ ($10^3$) $^c$ | M$_{n(GPC)}$$^d$ ($10^3$)/PDI |
|---|---|---|---|---|---|---|---|
| 1 | Bu$_4$NBr | 50:1:3 | 20 | 40 | 84 | 4.1 | 2.7/1.2 |
| 2 | Bu$_4$NBr | 50:1:3 | 20 | 25 | 86 | 4.3 | 3.6/1.1 |
| 3 | Bu$_4$NBr | 50:1:3 | 40 | 0 | 86 | 4.3 | 3.8/1.1 |
| 4 | Bu$_4$NBr | 50:1:3 | 68 | −10 | 79 | 3.9 | 3.6/1.2 |
| 5 | Bu$_4$NBr | 100:1:3 | 20 | 0 | 40 | 3.9 | 4.0/1.2 |
| 6 | PPNCl | 100:1:3 | 20 | 0 | 55 | 5.4 | 3.8/1.2 |
| 7 | Ph$_4$PCl | 100:1:3 | 20 | 0 | 46 | 4.5 | 3.6/1.2 |
| 8 | (Oct)$_4$NBr | 100:1:3 | 20 | 0 | 66 | 6.5 | 5.7/1.1 |
| 9 | (Oct)$_4$NBr | 100:1:5 | 22 | 0 | 74 | 7.4 | 5.7/1.1 |
| 10$^e$ | (Oct)$_4$NBr | 100:1:5 | 45 | 0 | 86 | 8.5 | 8.0/1.1 |
| 11 | (Oct)$_4$NBr | 100:1:5 | 45 | 0 | 93 | 9.2 | 8.6/1.1 |
| 12 | (Oct)$_4$NBr | 200:1:5 | 68 | 0 | 49 | 9.7 | 7.0/1.1 |
| 13 | (Oct)$_4$NBr | 200:1:10 | 45 | 0 | 79 | 16 | 10.0/1.1 |
| 14 | (Oct)$_4$NBr | 300:1:15 | 72 | 0 | 73 | 22 | 11.2/1.2 |
| 15 | (Bu$_4$N)$_2$CO$_3$ | 20:1:3 | 15 | 0 | 90 | 1.8 | 1.7/1.1 |
| 16 | (Bu$_4$N)$_2$CO$_3$ | 25:1:3 | 15 | 0 | 91 | 2.3 | 2.4/1.1 |
| 17 | (Oct$_4$N)$_2$CO$_3$ | 100:1:5 | 15 | 0 | 96 | 9.5 | 7.3/1.2 |

$^a$All polymerizations were carried out in 20 mL glass schlenk tube with rotaflo stopcocks under argon atomosphere in bulk conditions
$^b$Determined from $^1$H NMR
$^c$Calculated based on the formula: M$_{n(theo)}$ = 99 (DP$_{target}$) × (conversion %)
$^d$Determined by GPC in THF with polystyrene standard
$^e$Reaction was done using toluene as solvent, volume ratio of monomer: solvent = 3:1

with 3 equivalent of TEB resulted in 84% monomer conversion after 20 h. The sample was purified by precipitation in cold methanol and characterized by $^1$H NMR spectroscopy. Then a systematic study was conducted using TEB in excess for the polymerization of GA, the results of which are presented in Table 1.

As shown in Supplementary Figure 2a, the signals between 3.29 and 3.47 ppm correspond to the methylene protons (-CH$_2$-N$_3$) attached to the azido group; after the ring-opening of the epoxide, the peaks corresponding to methylene (-CH$_2$-O-) and methine (-CH-O-) protons move downfield and appear at 3.56–3.78 ppm. Such peak assignments were unambiguously confirmed by 2D NMR analysis (HSQC) (Supplementary Figure 2b). Moreover, the absence of proton signals in the olefin region (5.5–6.5 ppm) indicates that no elimination occurs during the anionic polymerization.

A further analysis by GPC indicates that the obtained PGA exhibits a narrow polydispersity but a lower molar mass than the expected value and a small bump in the high molar mass region (Fig. 3a). Such GPC traces are indicative of the occurrence of transfer reactions during the anionic polymerization of this substituted epoxide. There are two major possible pathways for such transfers to occur during PGA synthesis: (a) transfer to monomer (b) and/or transfer to polymer which both would lead to the formation of branched polymers of high molar mass as shown in Fig. 4. In order to obtain the evidence of transfer, the obtained polymer was submitted to MALDI-TOF analysis and the results are shown in Supplementary Figure 3a. In support to the GPC traces two distribution profiles are detected. The main distribution centered at 2584 m/z matches exactly with the expected structure (vide post for the interpretation). Another distribution centered at 3905 m/z does not include the same

terminal group (HBr) as the former one. Due to the occurrence of transfer reactions implying the displacement of an azido group from the polymer backbone by a growing alkoxide (Fig. 4, path b), the molar masses of this second population tend to double and the terminal group characterized by MALDI was found to be $C_3H_5OBr_2$. Indeed, the peaks of this minor population could be accounted for as: $99.09n + 217 + 23$, where 99.09, 217, and 23 correspond to the molar mass of GA, $C_3H_5OBr_2$ and Na, confirming that branching occurred via transfer to polymer. In addition, there is also one minor population encircled with a mass value of 2835 m/z (Supplementary Figure 3a, inset) which is found to have $C_3H_5OBr$ as terminal group. This corresponds $[99.09n + 137 (C_3H_5OBr) + 23(Na)]$ to polymer chains terminated with an epoxide end group due to transfer to monomer as

shown in Fig. 4 (path a). We demonstrate through this first series of experiments that boron-ate complexes that we used were effective at bringing about ROP of GA, although some transfer reactions to polymer and to monomer could not be avoided. To suppress the above-described chain transfers, polymerizations were conducted in a second step at lower temperatures such as 25 ℃, 0 ℃, and −10 ℃. It was observed that at 0 ℃ the polymerization proceeded smoothly with a minimum of transfer reactions as evidenced by GPC analyses and $M_{n(GPC)}$ values which were very close to the theoretical ones (entry 3, Table 1, Fig. 3b). Lowering further the temperature to −10 ℃ significantly slowed down the polymerization (entry 4, Table 1). Other initiators such as PPNCl, $Ph_4PCl$, and $Oct_4NBr$ were also screened at 0 ℃. The results (entry 5–8, Table 1) obtained showed that $Oct_4NBr$ was the most efficient initiator in terms of rate and control over the polymerization (entry 8, Table 1), the PGA polymer samples obtained using other initiators exhibiting deviation of their molar mass from the expected values. The ate complex derived from TEB and $Oct_4NBr$ indeed brought about a "living"/controlled polymerization of GA and the bulkiness of the $Oct_4N^+$ cation effectively prevented the transfer to the monomer and to the chain and thus the displacement of the azido functions.

The GPC analysis shows a monomodal distribution (Fig. 3), and the obtained $M_{n(GPC)}$ value (5.7 Kg/mol) indicates a value close to the theoretical one. Moreover, the absence of transfer reactions was verified by MALDI-TOF characterization, with only one distribution centered at 4952 detected (Fig. 5). All peaks in the main population appeared at $99.09n + 81 + 23$ with peak to peak mass difference of 99.09 for one GA unit, where 81 and 23 are the molar mass of HBr and Na. One small population showing 28 mass units less than the main population (Fig. 5, inset) and corresponding to the elimination of a nitrogen molecule during characterization is observed as it is for azide-containing polymers during such analysis[14].

Under these optimized conditions, PGA could thus be prepared in a controlled manner up to molar masses in the range of 11 kg/mol (entry 10 to 14) (Supplementary Figure 4) through either extending polymerization time or charging more TEB into the system. However, one should keep in mind that too high the concentration of TEB in the polymerization could in the meantime induce transfer reactions by enhancing the leaving

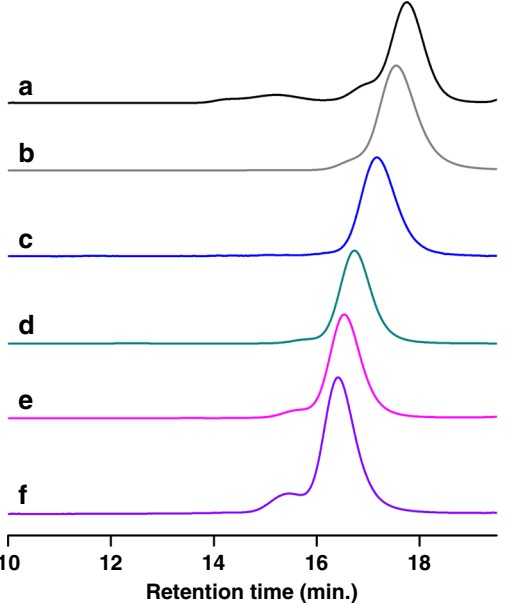

**Fig. 3** Representative GPC traces of PGA samples from Table 1. **a–f** Correspond to the GPC traces of the PGA samples of entries 1, 3, 8, 11, 13, and 14, respectively

**Fig. 4** Transfer reactions occurring during AROP of GA: Path a: transfer reactions occurring by displacement of an azido group from the monomer leading to end groups fitted with an epoxide (transfer to monomer). Path b: displacement of an azido group from the polymer backbone by a growing alkoxide (transfer to polymer). Both types of transfer reactions result in the formation of branched polymers

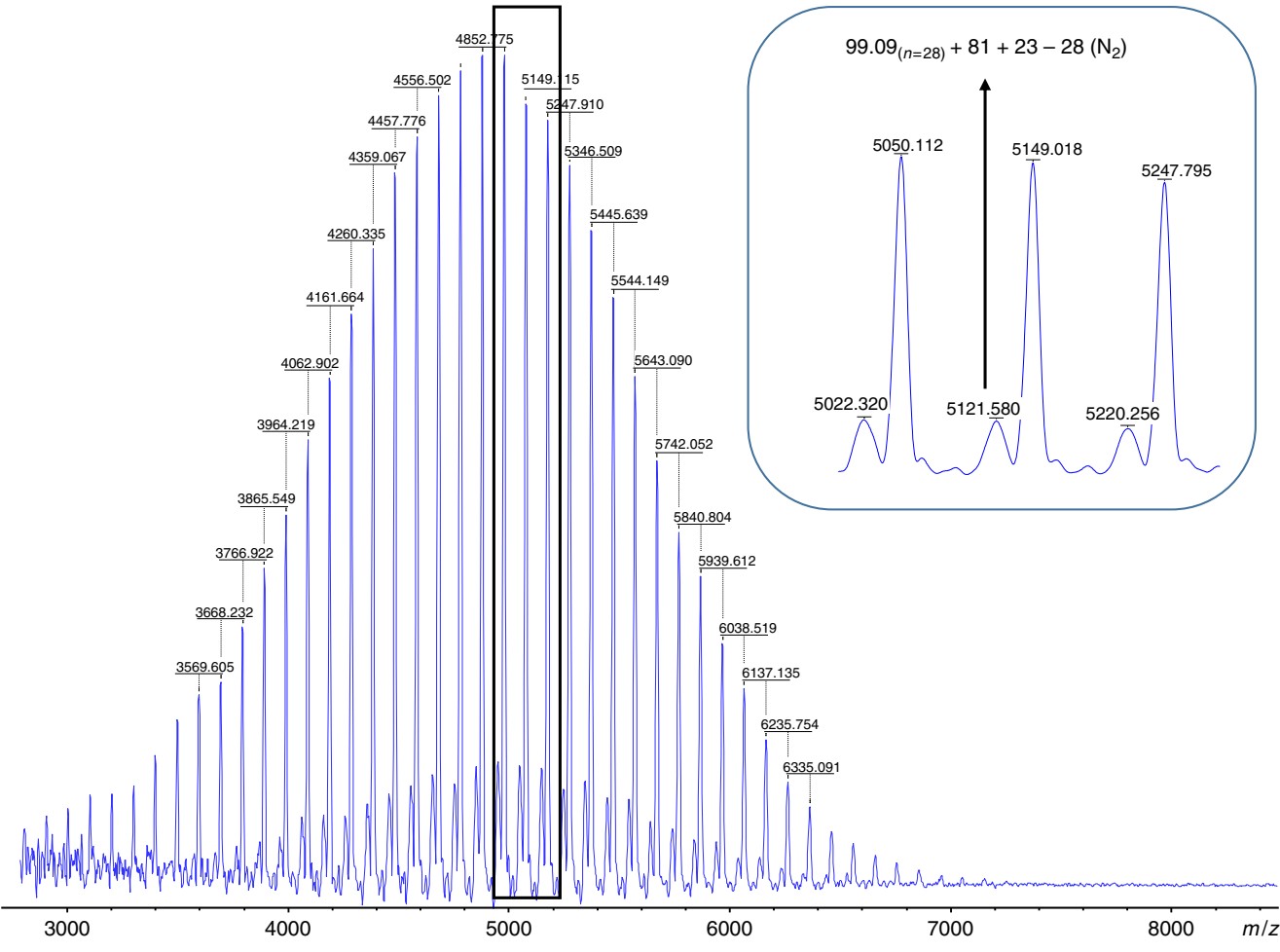

**Fig. 5** MALDI-TOF characterization result of the PGA sample corresponding to entry 8, Table 1. The main population corresponds to PGA chains carrying bromo (Br) and hydroxyl (OH) end groups [$99.09_n + 81 + 23$] with a peak to peak mass difference of 99.09 for one GA unit, and where 81 and 23 are the molar mass of HBr and $Na^+$. One small population showing 28 mass units less than the main population corresponds to the elimination of $N_2$ during analysis

ability of pendent azido functions. This can be attributed to the interaction of TEB with azido group as indicates the NMR characterization of the GA/TEB mixture which shows a downfield shift of the azido methylene protons (see Fig. 2, inset). Due to the dual role played by TEB which was found to interact with both the oxygen and the azido function of GA and activate them, attempts to get polymers of higher molar mass were not effective (Fig. 3f); In these cases, another population of higher molar mass was detected in GPC traces due to the occurrence of transfer reactions.

In lieu of TEB other Lewis acids were tried as a means to generate other types of ate complexes with different activities. We in particular attempted to utilize triisobutyl aluminium (TiBA), a Lewis acid described by Deffieux and Carlotti as effective at controlling the polymerization of functional epoxides such as epichlorohydrin in the presence of anionic initiators[7,14–16]. Unfortunately no polymerization of GA occurred in the presence of TiBA, nor in the presence of tris (pentafluorophenyl) borane under conditions described in the previously mentioned refs. [7,14–16]. Since the ate complex formed by the stoichiometric mixture of tetraoctylammonium bromide failed to ring-open GA, we then checked the ability of TiBA to activate GA. To our surprise and unlike the case of TEB we observed that TiBA reacts spontaneously with GA and triggers its ring opening in an uncontrolled manner (Supplementary

Figure 5). Addition of TiBA to the reaction medium containing GA (ratio = 1:1) indeed leads to an immediate color change, indicative of the reaction of the epoxide moiety. Actually, both [1]H NMR and [13]C NMR characterizations of the reaction medium shows the disappearance of the monomer and its ring opening; the GPC analysis showing the formation of ill-defined oligomers (See Supplementary Figure 6 and inset). Unlike TEB, TiBA could therefore not be successfully used in the case of GA as an ate complex forming Lewis acid and as GA activator.

We then focused our attention to the direct synthesis of hydroxyl-terminated telechelics of PGA (Supplementary Figure 7) through this method as they are a known application in the rocket fuel formulations. To this end, tetrabutylammonium and tetraoctylammonium carbonate salts ($(R_4N)_2CO_3$, R=Bu and Oct) were synthesized by treating the corresponding tetraalkylammonium hydroxide with carbon dioxide. Using this initiator, PGA-diols of molar mass 1700, 2400, 7300 g/mol (entry 15–17, Table 1) were successfully synthesized (For GPC traces of PGA-diol see Supplementary Figure 8) and the MALDI-TOF analysis confirmed the presence of PGA structure with two hydroxyl end groups (PGA-diol) (Supplementary Figure 3b). For instance, MALDI-TOF data displays two populations where the signal centered at m/z 2856 [99.09 (28) + 62 ($H_2O + CO_2$) + 23] corresponds to PGA-diol which has a carbonate unit embedded in the polyether chain arising from the carbonate initiator.

The $^1$H signals at 4.9 ppm and 4.13–4.33 pm correspond to the methine ($H_c$) and methylene ($H_d$) carbon linked to carbonate unit (when X=CO$_3$) indicates the presence of carbonate linkage (Supplementary Figure 9a) and further it was supported by the IR which shows a stretching frequency at 1747 cm$^{-1}$ corresponding to linear carbonate (See Supplementary Figure 10a). The mass signal at 2812 correspond to the PGA-diol without carbonate linkage which could be due to the loss of CO$_2$ during initiating step (Supplementary Figure 7). In any case the dihydroxy-ended structure was further confirmed by the reaction of PGA-diol (M$_{n\,(GPC)}$ = 2400 g/mol, entry 16, Supplementary Figure 9b) with an excess of phenylisocyanate with a catalytic amount of dibutyltin dilaurate (DBDTL). The $^1$H NMR characterization of the latter diurethane-modified PGA displayed the integral ratio of 2:2 for H$_a$ and H$_b$ which clearly supports the presence of two hydroxyl groups at the PGA chain ends [See Supplementary Figure 9b]. In addition, the hydroxyl value (51 mg KOH/g) and the functionality (2.18) were determined for the PGA-diol using the ASTM D1957-86 method[17] which shows a good agreement with the NMR values. Moreover, the polycondensation of the above PGA-diol with hexamethylene diisocyanate (1.1 equiv) was attempted using DBDTL as catalyst in DMF at 80 °C for 12 h; the polyurethane obtained exhibits a number average molar mass of 12.5 Kg/mol and no population corresponding to the double molar mass of PGA-diol, which further confirms the telechelics nature of the PGA-diol (See Supplementary Figure 10b for IR data and Supplementary Figure 11 for the GPC traces).

**Copolymerization of GA and CO$_2$.** Encouraged by the above homopolymerization results, we investigated the possibility of copolymerizing GA with CO$_2$ to produce carbonate-containing poly(glycidyl azide) that would then present the advantage of being degradable. Since GA is an analog of epichlorohydrin with its electron withdrawing group, it was thought that it could suffer from a selectivity problem and yield cyclic carbonates as main products of its copolymerization with carbon dioxide[18,19]. Indeed upon carrying out such copolymerization at 60 °C with ROH/P$_4$ as initiator in the presence of TEB [ratio of monomer: ROH/P$_4$: TEB = 100:1:2], under conditions similar to those of PO and CO$_2$[10], only cyclic carbonates of GA were produced as sole product in high yield (entry 1, Table 2). However upon lowering the

temperature to 25 °C the desired poly(glycidyl azide carbonate) (PGAC) was obtained with a 72% selectivity (M$_{n(GPC)}$ = 5.3 Kg/mol, entry 2, Table 2). To improve further the selectivity in favor of linear polycarbonate other onium salts such as PPNCl, Oct$_4$NBr, Bu$_4$NCl, and Bu$_4$NN$_3$ were tested for such copolymerizations. Cations of large size favor the formation of cyclic carbonates, and anions with higher nucleophilic character promote formation of linear carbonates in the following order N$_3^-$ > RO$^-$ > Cl$^-$ > Br$^-$ (entry 3–6, Table 2). This is consistent with previous works which show that the judicious choice of nucleophiles is crucial for the successful copolymerization of epoxide monomers containing electron-withdrawing groups with CO$_2$[18–20]. With Bu$_4$NN$_3$ used as initiator, excellent selectivity in favor of linear carbonate could be achieved; however the molar mass of PGAC obtained was 6.7 Kg/mol, which was lower than the theoretical value (entry 6, Table 2), indicating the existence of transfer; this could be evidenced by GPC analysis, where a bump was detected in the high molar mass region (Fig. 6b). The synthesis of PGAC thus also suffered both from the transfer reactions to the monomer and to the polymer, similarly to the synthesis of PGA as depicted in Fig. 4.

In an attempt to suppress transfer reactions, the ratio of TEB to initiator was reduced to 1.25: 1. Remarkably, under these conditions, PGAC of molar masses up to 8.2 Kg/mol, and in agreement to the expected value could be obtained (entry 7, Table 2). Moreover, the GPC trace displayed only a very small bump in the high molar mass region indicative of reduced transfers (Fig. 6c). $^1$H NMR analysis of the obtained PGAC (Supplementary Figure 12a) exhibits the characteristic signals at [5.01–5.03 ppm ($H_c$)] and [4.26–4.45 ppm ($H_b$)] corresponding to methine (-CH-O-CO$_2$) and methylene (-CH$_2$-O-CO$_2$-) protons of carbonate linkages and the signal at 3.41–3.72 ppm corresponding to the methylene protons ($H_a$) attached to azido function. The integral ratio of peaks of $H_c$, $H_b$, and $H_a$ (1: 2: 2) indicates alternating carbonate structure without formation of ether linkages, which was also supported by $^{13}$C NMR where there is no signal due to any etheral carbon (See Supplementary Figure 12b). A further reduction in the ratio of TEB to initiator (1:1) to minimize transfer reactions resulted in a significant loss of selectivity and increase of formation of cyclic carbonates (entry 8, Table 2).

**Table 2 Triethyl borane-assisted copolymerization of glycidyl azide with carbon dioxide[a]**

| Entry | Initiator | Activator | [M]:[I]:[A] | Temp (°C) | Time (h) | Selectivity[b] (%) | PC (%)[c] | Yield (%)[d] | M$_{n(theo)}$[e] (10$^3$) | M$_{n(GPC)}$[f] (10$^3$)/ PDI |
|---|---|---|---|---|---|---|---|---|---|---|
| 1 | ROH/P4 | TEB | 100:1:2 | 60 | 14 | 0 | - | (95)[g] | - | - |
| 2 | ROH/P4 | TEB | 100:1:2 | 25 | 14 | 72 | >99 | 53 | 7.5 | 5.3/1.1 |
| 3 | PPNCl | TEB | 100:1:2 | 25 | 14 | 50 | >99 | (83)[g] | - | 2.4/1.2 |
| 4 | Oct$_4$NBr | TEB | 100:1:2 | 25 | 14 | 0 | - | (99)[g] | - | - |
| 5 | Bu$_4$NCl | TEB | 100:1:2 | 25 | 14 | 90 | >99 | 79 | 11.3 | 6.5/1.1 |
| 6 | Bu$_4$NN$_3$ | TEB | 100:1:2 | 25 | 14 | >99 | >99 | 68 | 9.7 | 6.7/1.1 |
| 7 | Bu$_4$NN$_3$ | TEB | 100:1:1.25 | 25 | 14 | 93 | >99 | 63 | 9.0 | 8.2/1.1 |
| 8 | Bu$_4$NN$_3$ | TEB | 100:1:1 | 25 | 14 | 55 | >99 | 21 | 2.9 | 2.5/1.2 |
| 9 | Bu$_4$NN$_3$ | TEB | 100:1:1.25 | 10 | 14 | 76 | >99 | 43 | 6.1 | 2.3/1.2 |
| 10 | Bu$_4$NN$_3$ | TEB | 100:1:1.25 | 0 | 62 | 75 | >99 | 39 | 5.6 | 3.8/1.2 |
| 11 | Bu$_4$NN$_3$ | TEB | 100:1:1.25 | −5 | 120 | 94 | >99 | 60 | 8.7 | 7.0/1.1 |
| 12 | Bu$_4$NN$_3$ | TEB | 100:1:1.25 | −5 | 48 | 91 | >99 | 29 | 4.0 | 2.7/1.2 |
| 13 | Bu$_4$NN$_3$ | TEB | 100:1:5 | 25 | 14 | >99 | >99 | 94 | 14.3 | 4.6/1.2 |
| 14 | Bu$_4$NN$_3$ | TEB | 200:1:5 | 25 | 1 | >99 | >99 | 69 | 19.7 | 3.9/1.5 |

[a]All polymerizations were carried out in 50 mL autoclaves under 10 bar of CO$_2$
[b]Selectivity of linear carbonate calculated from IR spectra
[c]Determined from $^1$H NMR
[d]Calculated by gravimetry
[e]Calculated based on the formula: M$_{n(theo)}$ = 143 (DP$_{target}$) × (yield%)
[f]Determined by GPC in THF with polystyrene standard
[g]Values in the parenthesis corresponds to conversion determined from $^1$H NMR; ROH = monomethyl diethylene glycol

The next experiments were thus carried out at lower temperature to minimize transfer, the TEB to initiator ratio being fixed at 1.25:1. Accordingly, the temperatures of polymerization were reduced to 10, 0, and −5 °C respectively, the results of which being presented in entries 9–12, Table 2. It is observed that lowering of temperature has a small influence on the control over transfer reactions: the obtained molar mass still suffers from deviation with respect to the expected values even at −5 °C (entry 11, Table 2). Indeed, the GPC traces still displayed a bimodal distribution indicative of the occurrence of transfer reactions (Fig. 6d).

In order to better understand the mechanism of transfer during PGAC synthesis, the polymerization carried out at −5 °C was characterized at lower conversion (entry 12, Table 2, Fig. 6e). The $^1$H NMR analysis of the PGAC sample isolated clearly shows the occurrence of transfer reactions similarly to the case of PGA. For instance, the $^1$H NMR spectrum (Supplementary Figure 13) shows methylene (-CH$_2$-) signals at 2.83 and 2.95 and methine (-CH-) signals at 3.24 which indicate that the polycarbonate chains are terminated with epoxide end groups as proposed in Fig. 4 (path a). It is supported by MALDI-TOF data indicating that the minor peak encircled at mass 2983 m/z is found to have an end group corresponding to the molecular formula $C_7H_6N_6O_4$ (242.08, inset, Supplementary Figure 14), whereas all the populations of polycarbonates exhibit strictly alternating structures. As expected,

the signals of high intensity correspond to the PGAC initiated by azido function with a mass difference of 143 units and those of lower intensity such as 3014 m/z and 3000 m/z are due to the elimination of nitrogen atom during analysis.

To obtain PGAC with higher molar masses a ratio of TEB to Bu$_4$N$_3$ equal to 5 was tried (entry 13, 14). As expected, the yield of the PGAC increased; however, there was a decrease in the molar mass with a high polydispersity index due to transfer (Fig. 6f). It is evident that the synthesis of PGAC suffers from more transfers than the corresponding polyether synthesis even when lower temperatures and less TEB were used. In comparison to the homopolymerization of GA, it appears that the presence of the electro-withdrawing carbonate linkages increase the electrophilicity of the azido methylene carbon and thus facilitate transfer, which could not be totally suppressed under the copolymerization conditions.

**Co- and terpolymerization of GA with other epoxides and CO$_2$.** In order to take the advantage of the presence of azido functions, we investigated the incorporation of certain percentage GA units into other polyethers and polycarbonates through copolymerizations or terpolymerizations. Based on the optimum conditions found for GA copolymerization with CO$_2$, terpolymerization of CO$_2$, GA, and epoxides such as cyclohexene oxide (CHO) or propylene oxide (PO) were tried with a ratio of TEB to Bu$_4$NN$_3$ equal to 1:1 at room temperature. In general, a higher ratio of TEB to initiator (2:1) and temperature ≥ 60 °C was used for the copolymerization of CHO or PO with CO$_2$ for their polycarbonate synthesis[10]. Under such conditions, GA was found to undergo terpolymerization with epoxide monomers such as CHO or PO and CO$_2$ and yield the corresponding random polycarbonates (Table 3, entry 1–2).

For instance, the molar masses of the terpolymers obtained from CHO, GA, and CO$_2$ were close to the theoretical values, (entry 1, Table 3). Under the same reaction conditions GA was terpolymerized with PO and CO$_2$ giving a random polycarbonate in 41% yield with a good control of the molar mass. The symmetrical GPC distribution obtained in the above cases indicates that transfer reactions were negligible and not detected. (See Supplementary Figure 15 for the GPC traces). The $^1$H NMR spectra in Fig. 7 indicate the formation of a terpolymer involving PCHC, PPC, and PGAC and exhibiting an alternating polycarbonate structure and no ether linkage. Based on the intensities of their respective peaks, the composition of copolymers could be determined. According to Table 3, the feeding ratio of GA to CHO and to PO is 1: 4 and the composition of PGAC in PCHC and PPC backbone is found to be 1: 4.88 and 1: 0.49, indicating that the reactivity of the monomers follow the following order CHO ≥ GA > PO. Similarly, under the conditions of PGA synthesis, it was observed that GA could be incorporated in

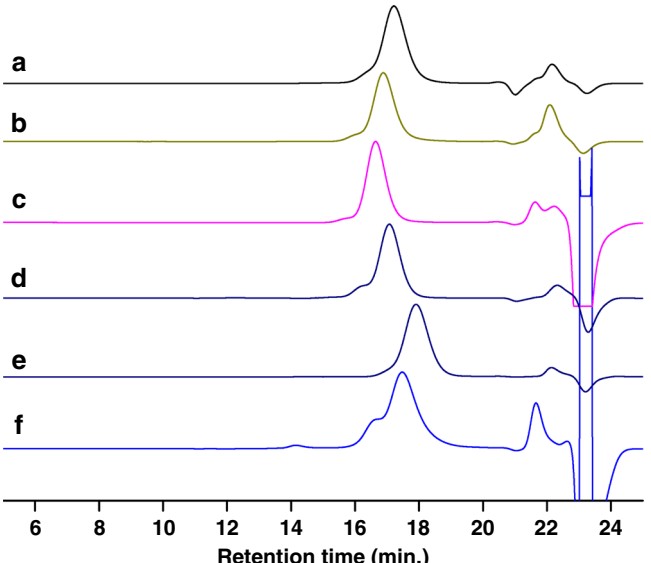

**Fig. 6** Representative GPC traces of PGAC samples from Table 2. **a**–**f** Correspond to the GPC traces of the PGAC samples of entries 2, 6, 7, 11, 12, and 14, respectively

**Table 3 Random terpolymerization of GA with epoxide monomers and CO$_2$ (entries 1 and 2) and copolymerization of GA with PO (entry 3)[a]**

| Entry | [M$_1$:M$_2$] | [I] | [M$_1$:M$_2$]:[I]:[A] | T (°C) | Selec. (%)[b] | PC (%)[c] | Tg (°C) | Yield (%)[d] | M$_{n(theo)}$[e] (10$^3$) | M$_{n(GPC)}$[f] (10$^3$)/PDI |
|---|---|---|---|---|---|---|---|---|---|---|
| 1 | GA:CHO | Bu$_4$NN$_3$ | 20:80:1:1 | 25 | >99 | >99 | 65 | 63 | 9.0 | 6.2/1.1 |
| 2 | GA:PO | Bu$_4$NN$_3$ | 20:80:1:1 | 25 | >99 | >99 | 24 | 41 | 4.5 | 6.0/1.1 |
| 3 | GA:PO | (Oct)$_4$NBr | 50:50:1:5 | 0 | - | - | -49 | 44 | 3.5 | 3.8/1.2 |

[a]Random polycarbonate (entry 1–2) were carried out in 50 mL autoclaves under 20 atm of CO$_2$ for 14 h; Random polyether synthesis (entry 3) was carried out in 20 mL glass schlenk tube for 2 h with rotaflo stopcock under argon atmosphere
[b]Selectivity (linear vs cyclic carbonate) calculated from IR spectra
[c]Polycarbonate content (PC) determined from $^1$H NMR
[d]Calculated by gravimetry
[e]M$_{n(theo)}$ calculated based on the formula: entry 1 = 142.2 (DP$_{target}$) × (yield%); entry 2 = 110.2 (DP$_{target}$) × (yield%); entry 3 = 78.5 (DP$_{target}$) × (yield%)
[f]Determined by GPC in THF with polystyrene standard

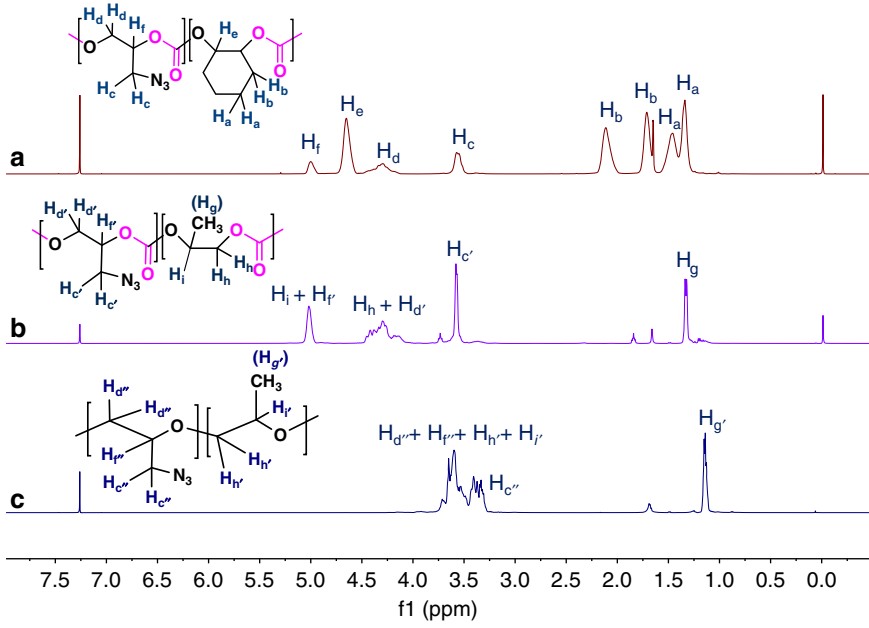

**Fig. 7** $^1$H NMR spectra of random copolymer samples of entry 1–3, Table 3 obtained through terpolymerization or copolymerization of GA with CHO(PO)/ $CO_2$ and with PO. **a**, **b** Indicate the random incorporation of GA in polycyclohexene carbonate and polypropylene carbonate backbone. **c** Displays the random incorporation of GA in poly(propylene oxide). Reaction conditions for **a**: GA/CHO /Bu$_4$NN$_3$/TEB = 20/80/1/1 and for **b**: GA/PO/Bu$_4$NN$_3$/ TEB = 20/80/1/1 at 25 °C and $CO_2$ (10 bar). For **c**: GA/PO/Oct$_4$NBr/TEB = 50/50/1/5 at 0 °C

the backbone of PPO through its copolymerization with PO (Fig. 7, Supplementary Figure 15). A higher reactivity of GA than that PO was also observed during copolymerization: starting from a feeding ratio of GA to PO equal to 1:1, the composition of PGA to PPO exhibited a ratio of 1.9:1.0 after random copolymerization. All the copolymers synthesized were subjected to DSC analysis and in all cases one unique glass transition temperature (T$_g$) was detected indicating the random structure of the copolymers formed (Table 3, Supplementary Figure 16).

**Derivatization of azido-carrying (co)polymers**. To demonstrate that the azido functions in PGA and PGAC can be easily modified by click chemistry without altering the PGA or PGAC chains a model compound, ethyl propiolate was used for the modification of the latter polymers through typical "click" conditions (Supplementary Figure 17). The complete transformation of azide into triazole one in both cases was confirmed by FTIR and NMR characterization, where the IR peak corresponding to azide (2093 cm$^{-1}$) completely disappeared, and new peaks appeared at δ 8.6 ppm corresponding to methine proton of the aromatic triazole ring (Supplementary Figure 18 and 19). The disappearance of the characteristic IR absorption at 2093 cm$^{-1}$ and the NMR integral ratios of peaks confirm the complete transformation. Moreover, the GPC analysis and MALDI-TOF characterization confirm the complete derivatization and integrity of polymer backbone under "click" conditions (Supplementary Figure 20 and 21). Lastly, the azido functions in PGA were reduced to primary amines using the Staudinger reaction (PPh$_3$, THF:H$_2$O) (Supplementary Figure 22 and 23). These primary amine carrying polyethers are currently investigated as $CO_2$ sorbent and the results will be reported soon.

## Discussion

We have demonstrated that glycidyl azide can be polymerized using a boron-ate complex as initiator under anionic conditions. It was found necessary to use an excess of the alkylboron Lewis acid to activate the GA monomer to trigger the polymerization.

Both homopolymerization of GA and its copolymerization with $CO_2$ were successful and produced PGA and PGAC with molar masses around 11 Kg/mol and narrow PDIs. Telechelic α,ω-dihydroxy PGA in the range of a few thousands molar mass could be directly synthesized for application such as propellant binder. In addition, thanks to the azido groups carried by GA its copolymerization with other epoxides provides a convenient method that afford polyethers and polycarbonates carrying reactive azido groups. Attempt has also been made to successfully modify these azide functions by click chemistry and produce other functional polyethers or polycarbonates.

## Methods

**Materials and characterization**. TEB in hexane solution (c = 1 M) was purchased from Aldrich and Acros and used as received. GA was prepared according to the literature procedure[21] and purified by distillation (not exceeding 35 °C since high temperature may lead to exothermic azide decomposition) under reduced pressure after stirring with calcium hydride for 3 days at room temperature. PO and CHO were purified by stirring with CaH$_2$ for 3 days followed by its distillation under reduced pressure. Tetraoctylammonium bromide (Oct$_4$NBr), Tetrabutylammonium chloride (Bu$_4$NCl) and tetrabutylammonium bromide (Bu$_4$NBr) were dried through azeotropic distillation using toluene as the solvent. Bis(triphenylphosphoranylidene)ammonium chloride (PPNCl) was purified by recrystallized from dichloromethane/diethyl ether mixture and dried in a vacuum oven at 60 °C overnight. CO$_2$ (99.995%) from Abdullah Hashim Industrial & Gas Co. was further purified by passing through a CO$_2$ purifier (VICI Co., US). All $^1$H and $^{13}$C NMR spectra were recorded on a Bruker AVANCE III-400 Hz instrument in CD$_2$Cl$_2$ and CDCl$_3$. GPC traces were recorded by VISCOTEK VE2001 equipped with Styragel HR2 THF and Styragel HR4 THF columns using THF (1 mL/min) as eluent. Narrow Mw polystyrene standards were used to calibrate the instrument. MALDI-TOF MS experiments were carried out by using trans-2-[3-(4-t-butyl-phenyl)-2-methyl2-propenylidene] malononitrile (DCTB) matrix (40 mg/mL) with sodium trifluoroacetate (2 mg/mL) as ionizing agent and polymer (2 mg/mL) in THF. From the above solutions, aliquots were taken in the ratio of 10:5:1 (matrix: polymer: Ionizing agent) for the analysis. DSC measurements were performed with a Mettler Toledo DSC1/TC100 under air. The samples were first heated from rt to 120 °C in order to erase the thermal history, then cooled to −90 °C, and finally heated again to 120 °C at a heating/ cooling rate of 10 °C min$^{-1}$.

**Representative procedure for homopolymerization of GA**. A pre-dried 20 mL glass schlenk tube composed of rotaflo stopcock and fitted with magnetic stirring bar was used to carry out this reaction. 55 mg of (Oct)$_4$NBr (0.1 mmol) was

dissolved in 1.0 g (10 mmol) of glycidyl azide in a 5 mL glass vial inside the glove box and then the whole solution is transferred to the glass schlenk tube under argon condition. The rotaflo was screwed tightly and 300 μL of triethyl borane (TEB, 1 M in hexane, 0.3 mmol) was taken in the head space of the schlenk tube and the outlet was closed with a glass stopper. The glass schlenk tube was taken out from the glove box and stirred at 0 °C for 10 min. Then the rotaflo was gently opened in order to add TEB into the monomer solution. The reaction mixture was stirred at the same temperature for 20 h, then it was quenched with few drops of 5% HCl aqueous solution. The quenched mixture was dissolved in THF (1 mL) and aliquots was taken for NMR and GPC analyses to determine the conversion and molar mass. The crude product was precipitated in methanol and the obtained PGA was characterized MALDI-TOF. The results were listed in Table 1 Entry 8.

The copolymerization of GA and PO was performed following a similar procedure as above.

**Representative procedure for GA and $CO_2$ copolymerization.** A 50 mL Parr reactor with magnetic stirrer dried at 120 °C for 6–8 h was taken into chamber of glove box, and dried under vacuum for 2–3 h. Then, the autoclave was transferred into glove box and placed inside the freezer (−30 °C) for 1 h. The copolymerization of $CO_2$ with glycidyl azide (GA) described below is taken from entry 6 in Table 2 as an example. The precooled autoclave was charged with $Bu_4NN_3$ (28 mg, 0.1 mmol) followed by the addition of 1 g of GA (10 mmol) to obtain a homogenous solution. To that monomer solution was added 200 μL of triethyl borane (TEB, 1 M in hexane, 0.2 mmol). Then the autoclave was sealed and taken out from the glove box. After charging $CO_2$ (10 bar), it was kept for stirring (250 rpm) at 25 °C for 14 h. The reactor was cooled down using ice-water bath for 15 min followed by the slow release of excess $CO_2$. The reaction mixture was then immediately quenched with dil. HCl (5% in methanol). The crude product was dissolved with $CHCl_3$ and then precipitated in cold methanol. The results were listed in Table 2 entry 7.

The terpolymerization of GA, $CO_2$ with CHO or PO was performed following a similar procedure as above.

## Data availability
The authors declare that the data supporting this study are available within the paper and its Supplementary Information File.

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

## Acknowledgements
This research work is supported by KAUST under baseline funding (BAS/1/1374-01-01).

## Author contributions
Y.G. and X.F. directed the investigations, and revised the manuscript. S.K.B. carried out all experiments and analyses, and wrote the draft. N.H. participated in the discussions and revised the manuscript.

## Additional information

**Competing interests:** The authors declare no competing interests.

