## [Peer Review File · Nature Communications]

Reviewers' comments:

Reviewer #1 (Remarks to the Author):

Gnanou et al.

"Direct Access to Poly(glycidyl azide) by Anionic Homopolymerization of Glycidyl Azide and to Various Azido-Containing Copolymers by its Copolymerization with Carbon Dioxide and Other Epoxides"

Gnanou et al. report the polymerization of glycidyl azide (GA) to polyether structures as well as polycarbonates. They show that the homopolymerization to PGA requires the addition of triethylborane as a weak Lewis acid catalyst. Surprisingly, the established tri-isobutyl aluminum catalyst does not lead to polymerization. This is rather unexpected, since Lynd et al. demonstrated that the structurally similar epichlorohydrin readily polymerizes upon addition of aluminum alkyls. The triethyl borane-mediated homopolymerization of glycidyl azide is an interesting achievement, however, I doubt that it is of sufficiently broad interest to be published in Nature Comm. I would wholeheartedly support publication in Polym. Chem. or Macromolecules.

Comments:

Page 2: "azidation" is the correct term, not "azidification"; page 18: what is "a broad polydispersity index" (dispersity may be high, but not broad);

SEC: Why were polystyrene standards employed? This calibration clearly leads to erroneous (i.e., apparent) molecular weights.

Page 6 and 12: Is there a mechanistic hypothesis, why TiBA does not activate polymerization of GA? TiBA activates the polymerization of other, rather similar epoxide monomer structures like ECH.

Page 9: Branching would also be supported (or disputed) by measurements of the intrinsic viscosity of PGA (if possible, online). In this case, the respective alpha-parameter would confirm the authors' structural conclusions.

For both syntheses: Is complete removal of TEB possible?

Copolymerization with PO: Is the resulting copolymer structure fully random?

Thermal properties of the polymers are crucial. Please add data, especially for the novel PGAC structure.

Reviewer #2 (Remarks to the Author):

The manuscript reports the preparation of poly(glycidyl azide) (PGA) through direct anionic ring opening polymerization of glycidyl azide (GA) in the presence of triethyl borane (TEB). This manuscript is composed of five relatively independent sections; preparation of homo-PGA, preparation of hydroxyl terminated telechelic PGA for polyurethane, copolymerization of GA with CO₂, copolymerization of GA/other epoxide/CO₂, and modification of azido-containing (co)polymers. A direct preparation of PGA from GA looks new. However, the five different works are presented in series without strong logical interconnectivity. This simple series of works make this manuscript too heavy and complex. The manuscript does not provide fundamental understanding on the reason why the direct polymerization of GA became possible in this case. The reason why the copolymerization with CO₂ should be done is also not well presented. Thus, the authors are advised to submit the manuscript in more concise form.

1. The authors better consider what the advances of their polymerization ingredients are, including plausible mechanism difference from other researcher's system. For example, why this ate anionic polymerization system is less sensitive to azido functionality? Why specifically "excess" amount of TEB is needed for the direct polymerization of GA? This better be assessed to fulfill the scientific interest of readers' of Nature Communication.

2. Although the background of the work for the preparation of homo- and telechelic PGA is presented, the need/reason/background for the introduction of CO₂ to PGA is not clear. Is

polycarbonate-azido better than PGA for solid rocket propellant? Otherwise, just another series of work that can be done by the same polymerization system?

3. Page 11 line 157. Although authors claimed that Oct4NBr was the most efficient initiator in terms of rate and control over the polymerization, it looks not clear because PDI was still high (~ 1.2), Mn(gpc) (5.7 x 10³ g/mol) was quite different from Mn(theo) (6.5, actually entry 5 looks better than entry 8, 3.9 versus 4.0), and conversion (66%) was still not that high.

4. Page 11 line 160. Although authors claimed the livingness of the polymerization, no evidence can be found in Table 1.

5. The general polydispersity of polymers in this study, PDI 1.2, looks not that narrow.

6. It looks like that the manuscript is not carefully prepared. Some example of such carelessness includes...

- Figure caption. Page 12 "Fig 4", However, all other figure caption in the main text is in the format of "Figure #". And in supporting information, all figure captions is now again "Fig #". Authors need to unify their presentation.

- Figure caption. Sometimes "Figure #.", sometimes "Figure #:". Same mistakes for "Fig S#." and "Fig S#:" in Supporting information.

- Figure caption. Sometimes punctuation at the end of the caption. Sometimes not. Same mistakes in both main manuscript and Supporting information.

- Figure caption. In Supporting information, sometimes A:..... B:....; but sometimes (A)... (B).... Authors need to unify their presentation.

- Table caption. Sometimes "Table #.", sometimes "Table #:"

- Scheme 1 and Scheme S2 are missing in pdf version of the manuscript, although this reviewer barely found those in original MS Word files. Authors better check their final submitted version of pdf.

- Figure S7B looks wrong. The chemical structure and integration numbers of protons indicate m = 1 (Page 13 line 208, Mn,gpc = 2400 g/mol?). Please check the chemical structure.

- Figures and tables should be placed in the order of their appearance in the main text. For example, in Page 8 line 127, author mentioned entry1 of GPC. However, the figure appeared far later than MADI-TOF figure.

- In addition, figure/scheme/tables should be inserted in the document after their first appearance in the main text. Scheme 1 should be placed after the paragraph of line 129 ~ 149.

- Some of experimental part (page 3 line 51 ~ page 4 line 57; page 5 line 82 ~ 86) is duplication of "Zhang, D.; Zhang, H.; Hadjichristidis, N.; Gnanou, Y.; Feng, X. Lithium-assisted copolymerization of co₂/cyclohexene oxide: A novel and straightforward route to polycarbonates and related block copolymers. *Macromolecules* 2016, 49, 2484-2492." It better be rewritten.

- Page 4 line 75 ~ 81 is almost a duplication of previous literatures. It also better be rewritten.

7. Page 6 Figure 1 caption. Authors should indicate that the figure is ¹H-NMR.

Reviewer #3 (Remarks to the Author):

The manuscript from Dr. Gnanou and coauthors is a very interesting and novel contribution in the field of anionic polymerization of functional epoxides. This is a solid paper, well written and the data are carefully discussed. I recommend publication after taking into account the following remarks and comments:

The copolymerization of GA with epoxides and CO₂ exhibits a quite low yield, do the authors have any explanation? What are the other products formed?

The modification of the polymers by click chemistry is a very interesting point and I would suggest to study it in greater detail. The modification degree should be determined from the NMR spectra to verify not only the disappearance of azide group but also the quantitative formation of the triazole moieties. Moreover, GPC analysis and MALDI would help to ensure that no main chain degradation is taking place.

Minor points are:

Scheme 1 and Scheme 2 are missing in the pdf version

Line 47 tetrabutylammonium

Line 70 was

Entry 7 table 1 Bu₄PCl, line 155 Ph₄PCl, experimental part Bu₄NCl for the same initiator?

Line 239 6.5 Kg/mol, table 2 entry 6 6.7 Kg/mol

Reviewers' comments:

Reviewer #1 (Remarks to the Author):

Gnanou et al.

"Direct Access to Poly(glycidyl azide) by Anionic Homopolymerization of Glycidyl Azide and to Various Azido-Containing Copolymers by its Copolymerization with Carbon Dioxide and Other Epoxides"

Gnanou et al. report the polymerization of glycidyl azide (GA) to polyether structures as well as polycarbonates. They show that the homopolymerization to PGA requires the addition of triethylborane as a weak Lewis acid catalyst. Surprisingly, the established tri-isobutyl aluminum catalyst does not lead to polymerization. This is rather unexpected, since Lynd et al. demonstrated that the structurally similar epichlorohydrin readily polymerizes upon addition of aluminum alkyls. The triethyl borane-mediated homopolymerization of glycidyl azide is an interesting achievement, however, I doubt that it is of sufficiently broad interest to be published in Nature Comm. I would wholeheartedly support publication in Polym. Chem. or Macromolecules.

Comments:

Page 2: "azidation" is the correct term, not "azidification"; page 18: what is "a broad polydispersity index" (dispersity may be high, but not broad);

The modifications suggested by the referee were introduced in the revised manuscript.

SEC: Why were polystyrene standards employed? This calibration clearly leads to erroneous (i.e., apparent) molecular weights.

Molecular weight values obtained by MALDI-ToF and by GPC using polystyrene calibration differ by less than 10%: this has been the reason for using GPC and polystyrene calibration for the determination of the molecular weight of PGA samples.

Page 6 and 12: Is there a mechanistic hypothesis, why TiBA does not activate polymerization of GA? TiBA activates the polymerization of other, rather similar epoxide monomer structures like ECH.

The polymerization of GA fails in presence of TiBA because of the chemical reaction between TiBA and GA possibly assisted by the azido functionality. To better understand the mechanism, TiBA and (Oct)₄N⁺Br⁻ were stoichiometrically mixed and the ate complex formed, (Br-AlR₃)⁻(Oct)₄N⁺, was utilized to initiate the polymerization of GA. No polymerization occurred under these conditions. When an excess of TiBA was added we could demonstrate that the latter was responsible for the true ring opening-opening of GA and not just its activation. In an independent experiment that consisted in mixing TiBA with GA we indeed clearly observed that TiBA reacted with GA at room temperature as indicates the spontaneous color change of the reaction medium. Both ¹H and ¹³C NMR of the aliquots show the disappearance of epoxide signals of GA and the formation of an oligomer, supported by GPC traces. This is very different from the case of epichlorohydrin (ECH) which is activated by TiBA but does not react with the latter Lewis acid. In the presence of an ate complex, as the one mentioned above, and with an excess of TiBA polymerization of ECH occurs by ring-opening initiated by the ate complex unlike the case of GA (See Fig 1 below for details).

Fig 1: ^1H NMR spectra displaying difference in the interactions of epoxide monomers GA and ECH with TEB and TiBA

Page 9: Branching would also be supported (or disputed) by measurements of the intrinsic viscosity of PGA (if possible, online). In this case, the respective alpha-parameter would confirm the authors' structural conclusions.

Branching is supported and evidenced by MALDI-ToF analysis which provide a clear information about the existence of transfer and then branching.

For both syntheses: Is complete removal of TEB possible?

TEB can be easily removed through precipitation of the polymer mixture in methanol. Unlike TIBA, TEB and any associated by-products formed at the time of quenching are easily removed by precipitation in methanol. This is further confirmed from the fact that there is no traces of TEB signal identified by the NMR spectra of purified polymers.

Copolymerization with PO: Is the resulting copolymer structure fully random?

The copolymerization of GA with either PO or CHO is random as evidenced by the DSC data obtained which indicate single Tgs for the copolymers obtained. Further, the Tg values obtained are in-between the values of their corresponding homopolymers. For instance, the Tg values of PCHC, PPC and PGAC are ~ 120 , ~ 35 and -2.5 $^\circ\text{C}$ respectively. The random polycarbonate

made of PGAC and PPC exhibits a T_g of 24 °C. Similarly, the random polycarbonate made of PGAC and PCHC exhibits a T_g of 65 °C. Additionally, PPO and PGA have T_g s of -60 and -44 °C and the random copolymer obtained from those exhibits a T_g of -49 °C which clearly shows that the obtained copolymer is random (See **Fig 2** below for details).

Fig 2: DSC curves displaying the T_g 's of **A:** P(GAC-co-PC); **B:** P(GAC-co-CHC); **C:** PGAC; **D:** P(GA-PO); **E:** PGA

Thermal properties of the polymers are crucial. Please add data, especially for the novel PGAC structure.

DSC data collected for all the newly synthesized polymers (PGA, PGAC, P(GAC-co-CHC), P(GA-co-PO) and the respective T_g values are added in the S.I. of the manuscript (See the Fig 2).

Reviewer #2 (Remarks to the Author):

The manuscript reports the preparation of poly(glycidyl azide) (PGA) through direct anionic ring opening polymerization of glycidyl azide (GA) in the presence of triethyl borane (TEB). This manuscript is composed of five relatively independent sections; preparation of homo-PGA, preparation of hydroxyl terminated telechelic PGA for polyurethane, copolymerization of GA with CO₂, copolymerization of GA/other epoxide/CO₂, and modification of azido-containing (co)polymers. A direct preparation of PGA from GA looks new. However, the five different works are presented in series without strong logical interconnectivity. This simple series of works make this manuscript too heavy and complex. The manuscript does not provide fundamental understanding on the reason why the direct polymerization of GA became possible in this case. The reason why the copolymerization with CO₂ should be done is also not well presented. Thus, the authors are advised to submit the manuscript in more concise form.

1. The authors better consider what the advances of their polymerization ingredients are, including plausible mechanism difference from other researcher's system. For example, why this anionic polymerization system is less sensitive to azido functionality? Why specifically "excess" amount of TEB is needed for the direct polymerization of GA? This better be assessed to fulfill the scientific interest of readers' of Nature Communication.

Scheme 1: Role of excess TEB for the GA Polymerization

The anionic polymerization of GA was successful in the presence of TEB because of the mild Lewis acidic character of TEB which was tolerant enough towards the chemically labile azido function and at the same time strong enough to activate monomer to drive the polymerization. This made the direct polymerization of GA possible with TEB. On the contrary this was not the

case with TiBA or more strong Lewis acids where undesired chemical reaction occurred between GA and the later Lewis acids.

As described in the above scheme, ate-complexes possess a weak nucleophilicity and basicity and because of these features undesired transfer reactions and proton abstraction from the monomer are minimized. Excess TEB is required for the GA polymerization to occur. TEB has thus a dual role: firstly, it forms an ate complex upon stoichiometric mixing with the initiating anion; secondly, its excess activates the monomer efficiently. (These descriptions are included in the manuscript).

2. Although the background of the work for the preparation of homo- and telechelic PGA is presented, the need/reason/background for the introduction of CO₂ to PGA is not clear. Is polycarbonate-azido better than PGA for solid rocket propellant? Otherwise, just another series of work that can be done by the same polymerization system?

The reason for the copolymerization of GA with CO₂ is included in the manuscript.

3. Page 11 line 157. Although authors claimed that Oct₄NBr was the most efficient initiator in terms of rate and control over the polymerization, it looks not clear because PDI was still high (~1.2), Mn(gpc) (5.7 x 10³ g/mol) was quite different from Mn(theo) (6.5, actually entry 5 looks better than entry 8, 3.9 versus 4.0), and conversion (66%) was still not that high.

The actual PDI obtained for the entry 8 was rechecked which is 1.1 and it is updated in the manuscript. The polymerization using Bu₄NBr (entry 5) is quite narrow however, it is less efficient to obtain high molecular weight as compared to Oct₄NBr.

4. Page 11 line 160. Although authors claimed the livingness of the polymerization, no evidence can be found in Table 1.

The livingness/"control" of the polymerization is supported by the control of the molecular weight molar of the samples obtained (the ratio of monomer to initiator and the conversion), the narrow PDIs measured (≤ 1.2 in Table 1) and the well-defined structure, which is supported by NMR and MALDI-ToF characterization.

5. The general polydispersity of polymers in this study, PDI 1.2, looks not that narrow.

The PDIs of PGA or PGAC obtained are in the range ≤ 1.2 which are comparable to values reported in the literature. Please refer to *Macromolecules* **2008**, *41*, 7058-7062 for polyether synthesis and *J. Am. Chem. Soc.* **2011**, *133*, 15191–15199 for polycarbonate synthesis. In addition narrow PDIs do not necessarily mean the livingness of the polymerization!

6. It looks like that the manuscript is not carefully prepared. Some example of such carelessness includes...

- Figure caption. Page 12 "Fig 4", However, all other figure caption in the main text is in the format of "Figure #". And in supporting information, all figure captions is now again "Fig #". Authors need to unify their presentation.

- Figure caption. Sometimes "Figure #.", sometimes "Figure #:". Same mistakes for "Fig S#." and "Fig S#:" in Supporting information.

- Figure caption. Sometimes punctuation at the end of the caption. Sometimes not. Same mistakes in both main manuscript and Supporting information.

- Figure caption. In Supporting information, sometimes A:..... B:....; but sometimes (A)...

(B)... Authors need to unify their presentation.

- Table caption. Sometimes “Table #.”, sometimes “Table #:”

All the captions including Figures, Tables and Schemes are unified.

- Scheme 1 and Scheme S2 are missing in pdf version of the manuscript, although this reviewer barely found those in original MS Word files. Authors better check their final submitted version of pdf.

- Figure S7B looks wrong. The chemical structure and integration numbers of protons indicate $m = 1$ (Page 13 line 208, $M_n, gpc = 2400$ g/mol?). Please check the chemical structure.

In Fig S7B, the ratio of protons H_a and H_b (2:2) indicates that the two ends of PGA are both functionalized by hydroxyl groups. “m” means here the polymerization degree, which is in fact 30 based on the 1H NMR data.

- Figures and tables should be placed in the order of their appearance in the main text. For example, in Page 8 line 127, author mentioned entry1 of GPC. However, the figure appeared far later than MADI-TOF figure.

Figure 4 and Table 1 are placed in proximity to make it convenient according to the main text.

- In addition, figure/scheme/tables should be inserted in the document after their first appearance in the main text. Scheme 1 should be placed after the paragraph of line 129 ~ 149.

Position of Scheme 1 has been changed according to the suggestion.

- Some of experimental part (page 3 line 51 ~ page 4 line 57; page 5 line 82 ~ 86) is duplication of “Zhang, D.; Zhang, H.; Hadjichristidis, N.; Gnanou, Y.; Feng, X. Lithium-assisted copolymerization of CO_2 /cyclohexene oxide: A novel and straightforward route to polycarbonates and related block copolymers. *Macromolecules* 2016, 49, 2484-2492.” It better be rewritten.

- Page 4 line 75 ~ 81 is almost a duplication of previous literatures. It also better be rewritten. Experimental part has been rewritten as mentioned.

7. Page 6 Figure 1 caption. Authors should indicate that the figure is 1H -NMR.

Correction done according to the suggestion.

Reviewer #3 (Remarks to the Author):

The manuscript from Dr. Gnanou and coauthors is a very interesting and novel contribution in the field of anionic polymerization of functional epoxides. This is a solid paper, well written and the data are carefully discussed. I recommend publication after taking into account the following remarks and comments:

The copolymerization of GA with epoxides and CO_2 exhibits a quite low yield, do the authors have any explanation? What are the other products formed?

The copolymerization of GA with CO₂ exhibits moderate yield due to their slow reaction rate at low temperature (≤ 22 °C). Any increment in temperature results in the formation of the corresponding cyclic carbonate. Efforts towards acceleration of polymerization rate by increasing TEB concentration indeed improved the yield; however it leads to undesired transfer as clearly described in the manuscript. Cyclic carbonate is the only by product obtained (or possible) during the polymerization.

The modification of the polymers by click chemistry is a very interesting point and I would suggest to study it in greater detail. The modification degree should be determined from the NMR spectra to verify not only the disappearance of azide group but also the quantitative formation of the triazole moieties. Moreover, GPC analysis and MALDI would help to ensure that no main chain degradation is taking place.

The derivatization of azido function into their corresponding triazoles in PGA and PGAC are clearly studied and their NMR data are added in the S.I. of the manuscript (See below Fig 3-7). The NMR interpretation showed that the polymers are quite stable during the derivatization as evidenced from their (polytriazoles) proton NMR integration. Moreover, the GPC and MALDI-ToF data of the polytriazoles of PGA and PGAC strongly confirms the stability and backbone integrity of the polymer. Lastly, the azido functions in PGA were reduced to primary amines using the Staudinger reaction (PPh₃, THF:H₂O). These primary amine-carrying polyethers are currently investigated as CO₂ sorbent and the results will be reported soon.

(All the data including GPC, MALDI-ToF are added in the S.I. of the manuscript).

Fig 3: ¹H NMR of Poly(glycidyl triazole)

Fig 4: ¹H NMR of Poly(glycidyl triazole carbonate)

Fig 5: GPC traces and MALDI Data of poly(glycidyl triazole)

Fig 6: GPC traces and MALDI Data of poly(glycidyl triazole carbonate)

Scheme S2: Synthesis of polyglycidyl amine

Fig 7: A: FTIR; B: ^1H NMR and C: ^{13}C NMR spectra of polyglycidyl amine

Minor points are:

Scheme 1 and Scheme 2 are missing in the pdf version

Line 47 tetrabutylammonium

Line 70 was

Entry 7 table 1 Bu₄PCl, line 155 Ph₄PCl, experimental part Bu₄NCl for the same initiator?

Line 239 6.5 Kg/mol, table 2 entry 6 6.7 Kg/mol

All the above corrections are updated.

Reviewers' comments:

Reviewer #1 (Remarks to the Author):

Since the other two reviewers support publication of this work and the authors have responded to all critical comments, I do now support publication of this work in Nature Comm. The authors made several important corrections in the manuscript, and it has been significantly improved.

Recommendation: Publish

Reviewer #2 (Remarks to the Author):

The manuscript looks better due to the revision based on the comments from the reviewers. However, some points are still not clear. This reviewer is still negative on the publication of the manuscript in Nature Communications.

Answer from the authors I)

3. The actual PDI obtained for the entry 8 was rechecked which is 1.1 and it is updated in the manuscript.

Comment I)

Authors cannot just change their data without enough reasons and grounds.

Answer from the authors II)

4. The livingness/"control" of the polymerization is supported by the control of the molecular weight molar of the samples obtained (the ratio of monomer to initiator and the conversion), the narrow PDIs measured (≤ 1.2 in Table 1) and the well-defined structure, which is supported by NMR and MALDI-ToF characterization.

5. In addition narrow PDIs do not necessarily mean the livingness of the polymerization!

Comment II)

In Table 1, $M_n(\text{gpc})$ values were quite different from $M_n(\text{theo})$ values. PDI values (1.1 ~ 1.2) still looks not that low. As author claimed, PDI value is even not enough to determine the livingness of the polymerization. Therefore authors should provide strong and solid sets of evidence if they want to claim the livingness of the system. It is author's obligation to provide enough evidence to claim something. It is also not clear what is the correlation between "well-defined structure" in their answer and the livingness of the polymerization. Therefore, evidence of livingness is still not enough in Table 1. Probably, authors need to show linear increase of conversion with time, continuous/linear increase of MW values with conversion, $M_n(\text{gpc}) \sim M_n(\text{theo})$ values, and low enough PDI values, etc....

Answer from the authors III)

1. These descriptions are included in the manuscript

2. The reason for the copolymerization of GA with CO₂ is included in the manuscript.

6.

- All the captions including Figures, Tables and Schemes are unified.

- Figure 4 and Table 1 are placed in proximity to make it convenient according to the main text.

- Position of Scheme 1 has been changed according to the suggestion.

- Experimental part has been rewritten as mentioned.

7. Correction done according to the suggestion.

Comment III)

Authors should pay enough attention before the submission of the manuscript, not after the submission and comments from the reviewers.

Answer from the authors IV)

6. In Fig S7B, the ratio of protons Ha and Hb (2:2) indicates that the two ends of PGA are both functionalized by hydroxyl groups. "m" means here the polymerization degree, which is in fact 30 based on the ^1H NMR data.

Comment IV)

Obviously, integration value of Hd (at 4.2 ppm) shown in Fig S7B is 2.15, which is almost same with the integration value of Ha (2.00) and Hb (2.01), indicating $m = 1$. If author want to claim that peaks in 3.2 ~ 3.8 ppm is also Hd, it should be indicated as such. Authors should be careful enough on their presentation.

Reviewer #3 (Remarks to the Author):

The comments and suggestions of the first report have been taken into consideration and data regarding the modification have been added. The problem with the pdf version and schemes 1 and 2 seems to persist. Since the remarks have been taking into account, I recommend publication

Answers to Reviewer (2) comments:

Answer from the authors I)

3. The actual PDI obtained for the entry 8 was rechecked which is 1.1 and it is updated in the manuscript.

Comment I)

Authors cannot just change their data without enough reasons and grounds.

Answer: It was a typo that occurred in the manuscript and hence it was corrected. The snapshot of the original data (entry 8, Table 1) is given below for reference. Also, MALDI-*tof* data of the same entry is provided in the manuscript in detail (Fig 4).

Fig 1: Snapshot displaying PDI 1.13 (entry 8, Table 1)

Answer from the authors II)

4. The livingness/"control" of the polymerization is supported by the control of the molecular weight molar of the samples obtained (the ratio of monomer to initiator and the conversion), the narrow PDIs measured (≤ 1.2 in Table 1) and the well-defined structure, which is supported by NMR and MALDI-*tof* characterization.

5. In addition narrow PDIs do not necessarily mean the livingness of the polymerization!

Comment II)

In Table 1, Mn(gpc) values were quite different from Mn(theo) values. PDI values (1.1 ~

1.2) still looks not that low. As author claimed, PDI value is even not enough to determine the livingness of the polymerization. Therefore authors should provide strong and solid sets of evidence if they want to claim the livingness of the system. It is author's obligation to provide enough evidence to claim something. It is also not clear what is the correlation between "well-defined structure" in their answer and the livingness of the polymerization. Therefore, evidence of livingness is still not enough in Table 1. Probably, authors need to show linear increase of conversion with time, continuous/linear increase of MW values with conversion, $M_n(\text{gpc}) \sim M_n(\text{theo})$ values, and low enough PDI values, etc....

Answer:

The experiment corresponding to entry 8 of the Table 1 of the manuscript was repeated and aliquots were withdrawn from the reaction medium with regular intervals to monitor the variation (a) of conversion with time, (b) and molar mass with conversion; as shown below both conversion and molar mass increase linearly with time and conversion, respectively.

Entry	Time (h)	Conversion (%)	M_n (GPC) $\times 10^3$ /PDI	$M_n \times 10^3$ (Theoretical)
1	1	8.5	0.51/1.40	0.84
2	2	14.3	0.90/1.37	1.38
3	3	18.5	1.36/1.37	1.83
4	4	21.4	1.78/1.37	2.11
5	6	32.3	2.70/1.32	3.19
6	8	38.3	3.56/1.26	3.79
7	10	42.2	3.98/1.26	4.17
8	14	55.3	4.50/1.25	5.47
9	18	66.5	5.40/1.17	6.58

Fig 2a: Displaying linear increase of Conversion with Time

Fig 2b: Displaying linear increase of M_n with conversion

Fig 3: GPC traces displaying increase in M_n (Table above, entries 1-9)

These results confirm that the anionic polymerization of glycidyl azide proceeds under “living”/controlled conditions. Both in the manuscript and in the answers to the reviewers’ queries, we managed not to claim that the anionic polymerization of glycidyl azide is truly “living”, meaning that it is truly free of any termination and transfer reactions whatever the conditions used. We instead mentioned that anionic polymerization of glycidyl azide proceeds under “living”/controlled conditions, meaning that in the range of molar masses investigated –typically below 11,000g/mol- we could achieve a good control of the sample molar masses and could not detect any termination or transfer reactions.

Answer from the authors III)

1. These descriptions are included in the manuscript
 2. The reason for the copolymerization of GA with CO₂ is included in the manuscript.
 - 6.
- All the captions including Figures, Tables and Schemes are unified.
 - Figure 4 and Table 1 are placed in proximity to make it convenient according to the main text.

- Position of Scheme 1 has been changed according to the suggestion.
 - Experimental part has been rewritten as mentioned.
7. Correction done according to the suggestion.

Comment III)

Authors should pay enough attention before the submission of the manuscript, not after the submission and comments from the reviewers.

Answer from the authors IV)

6. In Fig S7B, the ratio of protons Ha and Hb (2:2) indicates that the two ends of PGA are both functionalized by hydroxyl groups. "m" means here the polymerization degree, which is in fact 30 based on the ¹H NMR data.

Comment IV)

Obviously, integration value of Hd (at 4.2 ppm) shown in Fig S7B is 2.15, which is almost same with the integration value of Ha (2.00) and Hb (2.01), indicating m = 1. If author want to claim that peaks in 3.2 ~ 3.8 ppm is also Hd, it should be indicated as such. Authors should be careful enough on their presentation.

Answer: The chemical structure in Fig S7B is corrected as mentioned by the reviewer and updated in the supplementary information.

The corrected Fig is given below for the reviewer's kind reference.

Fig S6: ¹H NMR spectra of **A:** PGA-diol (entry 16, Table 1) and **B:** Urethane adduct of PGA-diol and phenylisocyanate